# Laminar specificity of the auditory perceptual awareness negativity: A biophysical modeling study

**Carolina Fernandez Pujol** [ORCID]*, **Elizabeth G. Blundon**¤, **Andrew R. Dykstra** [ORCID]*

Department of Biomedical Engineering, University of Miami, Coral Gables, Florida, United States of America

¤ Current address: Department of Medicine, Dalhousie University, Halifax, Nova Scotia, Canada
* cxf418@miami.edu (CFP); adykstra@miami.edu (ARD)

**Data Availability Statement:** All data and code required to reproduce the findings in this paper are available at: https://osf.io/9s3mq/?view_only=0ef80aeda9524c7092bbc7c639a620bf.

## Abstract

How perception of sensory stimuli emerges from brain activity is a fundamental question of neuroscience. To date, two disparate lines of research have examined this question. On one hand, human neuroimaging studies have helped us understand the large-scale brain dynamics of perception. On the other hand, work in animal models (mice, typically) has led to fundamental insight into the micro-scale neural circuits underlying perception. However, translating such fundamental insight from animal models to humans has been challenging. Here, using biophysical modeling, we show that the auditory awareness negativity (AAN), an evoked response associated with perception of target sounds in noise, can be accounted for by synaptic input to the supragranular layers of auditory cortex (AC) that is present when target sounds are heard but absent when they are missed. This additional input likely arises from cortico-cortical feedback and/or non-lemniscal thalamic projections and targets the apical dendrites of layer-5 (L5) pyramidal neurons. In turn, this leads to increased local field potential activity, increased spiking activity in L5 pyramidal neurons, and the AAN. The results are consistent with current cellular models of conscious processing and help bridge the gap between the macro and micro levels of perception-related brain activity.

## Author summary

To date, our understanding of the brain basis of conscious perception has mostly been restricted to large-scale, network-level activity that can be measured non-invasively in human subjects. However, we lack understanding of how such network-level activity is supported by individual neurons and neural circuits. This is at least partially because conscious perception is difficult to study in experimental animals, where such detailed characterization of neural activity is possible. To address this gap, we used biophysical modeling to gain circuit-level insight into an auditory brain response known as the auditory awareness negativity (AAN). This response can be recorded non-invasively in humans and is associated with perceptual awareness of sounds of interest. Our model shows that the AAN likely arises from specific cortical layers and cell types. These data help bridge the

**Funding:** This work was supported by the National Institute on Deafness and Communication Disorders under award number 1R21DC020295-01 (to ARD), the National Center for Advancing Translational Sciences under award number UL1TR002736 (to ARD via the University of Miami Clinical and Translational Science Institute), a seed grant from the Fondation Pour l'Audition (to ARD), and a McKnight Doctoral Fellowship from the Florida Education Fund (to CFP). The funders had no role in study design, data collection and analysis, decision to publish, or preparation of the manuscript.

**Competing interests:** The authors have declared that no competing interests exist.

gap between circuit- and network-level theories of consciousness, and could lead to new, targeted treatments for perceptual dysfunction and disorders of consciousness.

## Introduction

A central question of cognitive neuroscience is to identify which patterns of brain activity underlie conscious perception of sensory stimuli [1–3]. Much of what is known concerning this question comes from human neuroimaging studies that rely on non-invasive techniques such as electroencephalography (EEG), magnetoencephalography (MEG), and functional magnetic resonance imaging (fMRI) that can record human brain activity during perception and cognition. Such experiments have led to fundamental insight into the large-scale, network-level dynamics of conscious perception. However, despite recent experiments in animal models [4,5] and associated theoretical proposals [6–8], we still lack basic understanding of the cellular- and circuit-level brain dynamics that support conscious perception, particularly across species and sensory modalities.

This is likely due to inherent limitations in each neuroimaging method that hinder cellular- and circuit-level interpretation. In M/EEG, it has been estimated that 10,000–50,000 [9] pyramidal neurons must receive synchronous synaptic input such that the currents in their apical dendrites align in space and time to generate measurable evoked responses outside the head. Such currents in the apical dendrites of pyramidal neurons, which are the primary generators of M/EEG responses [10–14], are often represented by a single, equivalent current dipole. While the spatial accuracy of such dipoles is often reasonable, advanced methods are required for the accuracy and precision necessary to make empirical inferences at the cellular- and circuit-level [15]. In the case of ultra-high-field fMRI, while laminar specificity is sometimes possible [16,17], the BOLD signal is both temporally sluggish and only a proxy for neural activity. Even in human intracranial recordings, which are resolved in both time and space, electrode placement is solely determined by clinical considerations, limiting access to primary sensory and motor regions, particularly insofar as laminar inference is concerned [18,19].

Consequently, most of what is known about cellular- and circuit-level dynamics of perception- and consciousness-related activity comes from studies using invasive animal models. For example, it has been shown in mice that anesthesia disrupts the (normally strong) coupling between apical (near dendritic tuft) and perisomatic compartments of L5b pyramidal neurons. This coupling, which depends on input from the posteromedial thalamic nucleus to metabotropic receptors located between the apical and perisomatic compartments, enables $Ca^{2+}$ spikes in the apical compartment to elicit/amplify spiking at the soma in awake mice [20]. Additional experiments in mouse barrel cortex that are more directly relevant for the present study have shown that such active $Ca^{2+}$ processes in the apical compartment, which reflect both cortico-cortical feedback and thalamo-cortical loops, are causally involved in the perception of whisker deflections [4,5]. However, as far as we know, such findings have not been replicated in other species or modalities. In fact, what we do know about human dendrites is that they are longer and electrically more isolated (i.e., less coupled) between their apical and perisomatic compartments [21]. This would make apical $Ca^{2+}$ spikes less likely to trigger somatic $Na^+$ spiking, which in turn would render functional coupling between the apical and perisomatic compartments both less effective but more specific and computationally richer [22,23]. Clearly, more work is needed to bridge the gap between cellular- and network-level understanding of conscious perception.

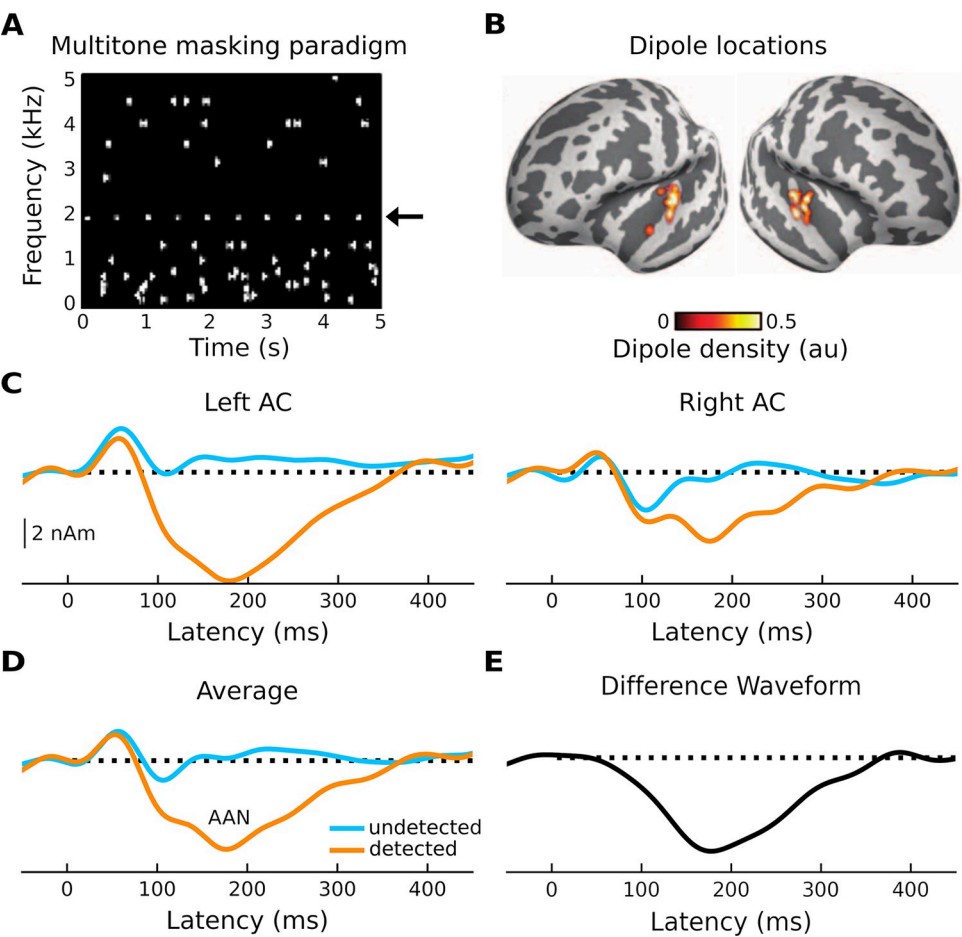

**Fig 1. Stimulation paradigm and evoked responses.** (**A**) Trial consisting of a sequence of standard tones (the target) embedded in a "cloud" of tones placed randomly in frequency and time (multitone masking). (**B**) Source locations of responses shown in C. (**C**) MEG activity of undetected (blue trace) and detected (orange trace) standard tones for both left and right auditory cortex. (**D**) Average of undetected and detected responses across hemispheres. (**E**) Difference waveform between the recorded response to the undetected and the recorded response to detected standard tones. Adapted from [26]. Negative is plotted down.

Here, we used biophysical modeling—the Human Neocortical Neurosolver (HNN) [14]—to examine the cellular- and circuit-level basis of an *auditory* evoked response associated with conscious perception of target sounds in noise (Fig 1A and 1B). HNN can be used to develop or test hypotheses about underlying circuitry that gives rise to evoked responses and has been previously applied to study the cell and circuit origin of auditory evoked responses [24]. HNN provides a canonical model of a neocortical column circuit that can be driven via two types of exogenous input (proximal and distal) and allows for one-to-one comparison between model results and source-localized M/EEG data. The specific response we sought to model is the auditory awareness negativity, or AAN [25–28]. The AAN, which arises from auditory cortex (AC), can be thought of as an auditory version of a perceptual awareness negativity (PAN) [29], a reliable M/EEG marker of perception-related activity that has counterparts in both the visual [30] and somatosensory [31,32] modalities. The PAN consists of a negative component arising between approximately 120 and 200 ms following stimulus onset in modality-specific cortex [29]. To improve our understanding of brain-behavior relationships as it pertains to auditory conscious perception, our aim was to generate a layer-specific hypothesis that

describes how a neocortical column circuit generates the AAN. The data used in our model was taken from previous work that examined MEG responses to both detected and undetected target-tone streams embedded in random, multitone maskers (Fig 1) [26]. Modeled components include an early positivity (P1), an early negativity (N1), and the AAN, which can be isolated as the difference between the response to detected and undetected target tones. The P1 and N1 were generated in response to both detected and undetected tones, but the AAN was only generated in response to detected tones. Our modeling results showed that the AAN can be accounted for by additional input targeting superficial layers, which results in a prolonged surface negativity—the AAN—and greater firing of L5 pyramidal neurons. These results are consistent with current cellular models of consciousness and conscious perception [6–8] that highlight the importance of active mechanisms in the apical dendrites of L5 pyramidal neurons [33,34].

## Results

### Biophysical model

MEG signals—i.e. magnetic fields generated inside the brain that can be measured outside the head—are known to arise from longitudinal intracellular currents predominantly from apical dendrites of spatially aligned pyramidal neurons [12,13]. These currents arise from synaptic inputs that target different parts of the cell. The same synaptic activity (and resulting intracellular currents) also results in buildup of extracellular charge that can be recorded outside the head as EEG.

The details of the biophysical model implemented by HNN are described elsewhere [14] (cf. Materials and methods). Briefly, it models the primary MEG currents with a reduced neocortical column circuit that includes both excitatory pyramidal and inhibitory basket neurons in cortical layers 2/3 and 5 (Fig 2) [14]. The circuit can be driven via two types of inputs to simulate event-related fields (ERFs) of interest: (i) proximal and (ii) distal inputs. Proximal inputs represent feedforward information flow; inputs arrive from the lemniscal thalamus and target the basal dendrites of pyramidal neurons and the somata of basket cells across layers. Net excitatory proximal inputs result in current flow up the pyramidal neuron dendrites, towards the cortical surface, but can also result in current flow down the PN dendrites via somatic inhibition. Distal inputs, on the other hand, represent feedback information flow; inputs arrive from either cortico-cortical projections or the non-lemniscal thalamus, and target the apical dendrites of pyramidal neurons (across layers) and the soma of basket cells in layer 2/3. Net excitatory distal input results in current flow down the dendrites, towards the white matter surface. These inputs would result in magnetic fields that point around the primary current dipole axis, with opposite field patterns for proximal and distal inputs. Importantly, HNN's modeled magnetic fields are in the same units (nAm) as source-localized MEG signals, permitting one-to-one comparison of model and data.

### Evoked responses

We first modeled the response to undetected target tones, which consists of a positivity at around 50 ms (most likely a mixture of the auditory Pa and P1) that represents an auditory stimulus reaching the primary AC, and a small negativity at around 100 ms that can be thought of as an N1 arising from auditory association cortex (Fig 3A). To reproduce this morphology, we began by hand-tuning the timing and strengths of each driving input while visually comparing the shape of the average simulated ERF (dark blue) to that of the recorded one (light blue). Simulated individual trials (n = 10) are shown in gray. The fit was then optimized using HNN's optimization tool, resulting in an $R^2$ value of 0.57 and a root mean-squared error

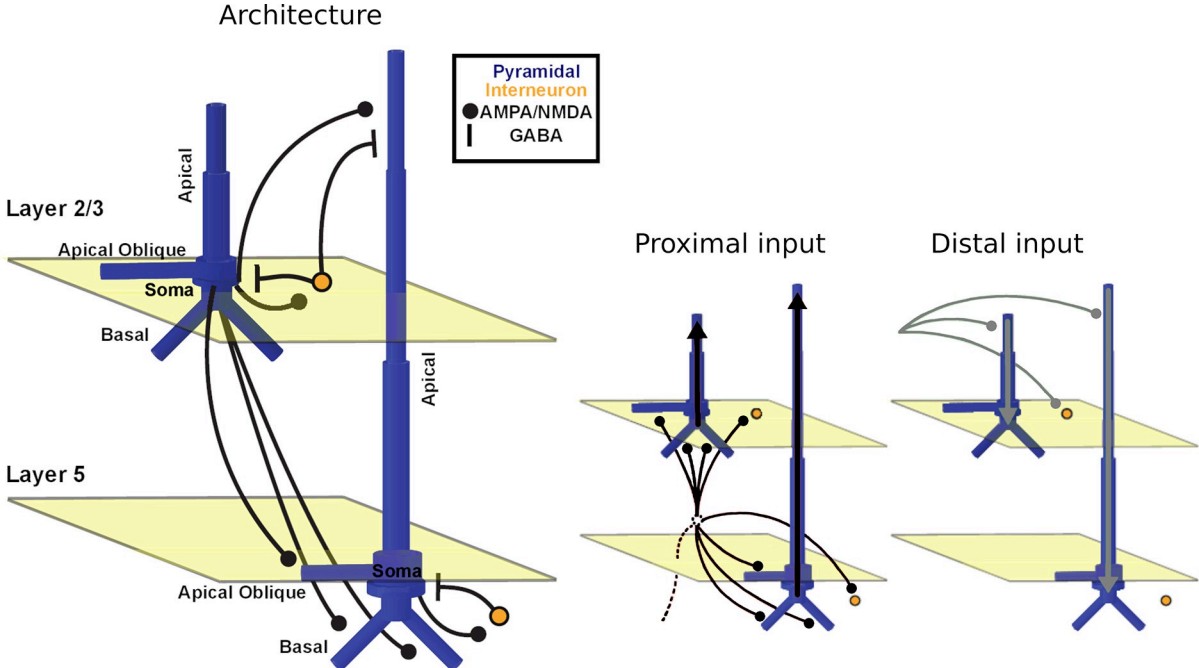

**Fig 2. The Human Neocortical Neurosolver.** Architecture / underlying connectivity of HNN model cortical column and inputs (proximal input in black and distal input in gray) [14].

(RMSE) value of 0.61 nAm. The final, optimized model contains two driving inputs: a proximal (i.e., feedforward) input at 47 ms and a distal (i.e., feedback) input at 84 ms. We then modeled the response to detected target tones, which contain an additional negativity between approximately 100 and 300 ms—the AAN (Fig 3B). To achieve a close fit for this response, we held the parameters of the first two inputs fixed, and hand-tuned/optimized the parameters of a second, later distal input (169 ms), which resulted in an $R^2$ value of 0.95 and an RMSE of 0.63 nAm. This second, broader distal input results in driving current down the dendrites, and closely reproduces the AAN, which reflects perceptual awareness of the target tones [25,26].

The first question we wanted to address was whether the AAN arises from input to specific cortical layers. The laminar-specific contributions to the simulated waveforms (Fig 3C and 3D) show (i) that the biphasic response to undetected targets arises from activity in both layers 2/3 (green, positive-going) and 5 (negative-going), and (ii) that the difference between responses to detected and undetected target tones is mostly due to additional negative-going activity in layer 5 (magenta traces) (with only a small contribution from negative-going activity in layer 2/3). All inputs to the model and their corresponding parameters can be found in Fig 4.

## Local field potentials

In addition to the simulated current dipoles of Fig 3, which represent net intracellular current in the apical dendrites of pyramidal neurons and the main contributors to measured M/EEG fields [35], our biophysical model can also simulate local field potentials (LFPs). LFPs refer to electrical activity recorded in the extracellular space around neurons (and possibly other cell types) from within the cortex. Experimentally, LFPs can be measured across cortical layers using linear probes [36,37] inserted perpendicularly [38] into the cortex (as opposed to M/EEG, which record activity from outside the head). The invasive aspect of this approach makes access to LFPs in humans challenging (but see [18,39]), even more so in primary sensory regions like AC. The model column provided by HNN and simulated multielectrode array are

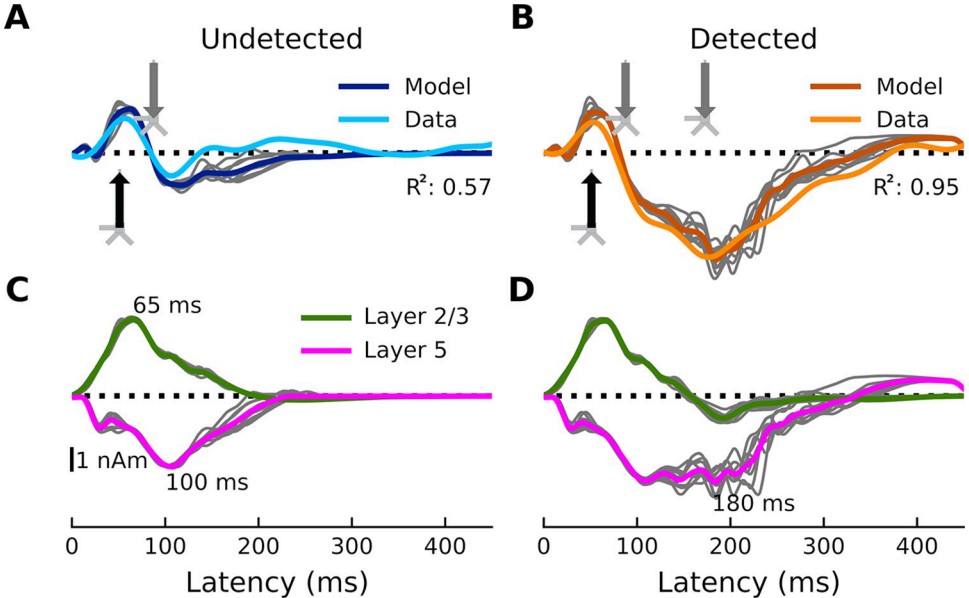

**Fig 3. Current dipoles and laminar profiles. (A)** HNN simulation for the average response to the undetected target tones (dark blue trace). Input spikes are sampled from a Gaussian distribution on each trial for a total of 10 trials (gray traces). A proximal input (47.8 ms) followed by a distal input (84.3 ms) drive the network. $R^2$ between empirical (light blue trace) and simulated data (dark blue) is 0.57 and RMSE is 0.61 nAm. **(B)** HNN simulation for the average response to detected target tones (dark orange traces). The same proximal and distal inputs used to model the response to undetected target tones, and an additional distal input (169.3 ms), drive the network. $R^2$ between empirical (light orange trace) and simulated data (dark orange) is 0.95 and RMSE is 0.63 nAm. **(C-D)** Laminar profiles for the responses to the undetected and detected target tones. The contribution of layer 2/3 (which reflects the longitudinal currents in layer-2/3 (L2/3) pyramidal neurons only) is plotted in green and that of layer 5 (which reflects the longitudinal currents in L5 pyramidal neurons only) in magenta. Note that the model traces in panels A and B reflect the aggregate of the green and magenta traces in panels C and D. The corresponding input parameter values are displayed in Fig 4. The network from which the simulated dipole activity arises consists of 60,000 cells.

shown in Fig 5A. Simulated LFPs to the undetected (Fig 5B) and detected (Fig 5C) target tones are plotted for a single trial. The traces show that the positivity at ~50 ms and the negativity at ~100 ms are associated with LFPs mostly in layer 2/3 (note that there is also a relatively small amount of synaptic activity in layer 5) but results in strong dendritic currents in both layer 2/3 (towards the cortical surface) and layer 5 (away from the cortical surface) (Fig 3C).

## Spiking activity

The second question we wanted to address was whether the AAN is correlated with cell output in specific cortical layers. In response to undetected target tones (Fig 6A), the model shows (i) brief spiking of a subset of L2/3 basket cells (open green circles), (ii) large but brief spiking of L2/3 pyramidal neurons (solid green triangles), and (iii) sustained basket-cell firing in layer 5 (open magenta circles) and (more so) layer 2/3 (open green circles); very little spiking is observed in L5 pyramidal neurons. In response to detected target tones (Fig 6B), the model shows the same initial spiking activity as for undetected target tones, but is followed by (i) increased, burst-type spiking activity in L5 pyramidal neurons, (ii) even more sustained firing of basket cells in layer 2/3, (iii) rhythmic firing of basket cells in layer 5, and (iv) slightly enhanced firing of L2/3 pyramidal neurons in later time windows. In answering the question posed earlier, this work is proposing a model that accounts for the AAN by the means of additional input to the superficial layer dendrites that leads to increased firing of L5 pyramidal neurons.

| Parameter | | | Undetected | | Detected | | |
|---|---|---|---|---|---|---|---|
| | | | Proximal ⬆ | Distal ⬇ | Proximal ⬆ | Distal ⬇ | Distal ⬇ |
| Input time (ms) | | | 47.815 | 84.275 | 47.815 | 84.275 | 169.273 |
| SD (ms) | | | 13.217 | 15.063 | 13.217 | 15.063 | 50.396 |
| Weight (μS) | Layer 2/3 Pyramidal | AMPA | 1.683 | 0.0107 | 1.683 | 0.0107 | 0.164 |
| | | NMDA | 2.762 | 0.03 | 2.762 | 0.03 | 0.0703 |
| | Layer 2/3 Basket | AMPA | 2.028 | 0.645 | 2.028 | 0.645 | 0.593 |
| | | NMDA | 0.916 | 0.988 | 0.916 | 0.988 | 0.175 |
| | Layer 5 Pyramidal | AMPA | 0.001 | 0.0005 | 0.001 | 0.0005 | 0.0464 |
| | | NMDA | 0.000999 | 0.0149 | 0.000999 | 0.0149 | 0.297 |
| | Layer 5 Basket | AMPA | 0.4751 | | 0.4751 | | |
| | | NMDA | 0.0614 | | 0.0614 | | |

**Fig 4. Driving inputs (and parameters) used to model the responses to the undetected and detected target tones.** The model column used was the calcium model column that consists of a more biologically accurate distribution of $Ca^{2+}$ channels on L5 pyramidal neurons compared to HNN's original Jones 2009 model column.

## Alternative models

Since net excitatory distal inputs result in driving current down the pyramidal neuron dendrites, one way to account for the AAN is to drive the model column with a distal input (as we demonstrated in the previous section). Given the latency of the PAN (approximately 100–300 ms after stimulus presentation), which is sufficiently long to have engaged cortico-cortical or

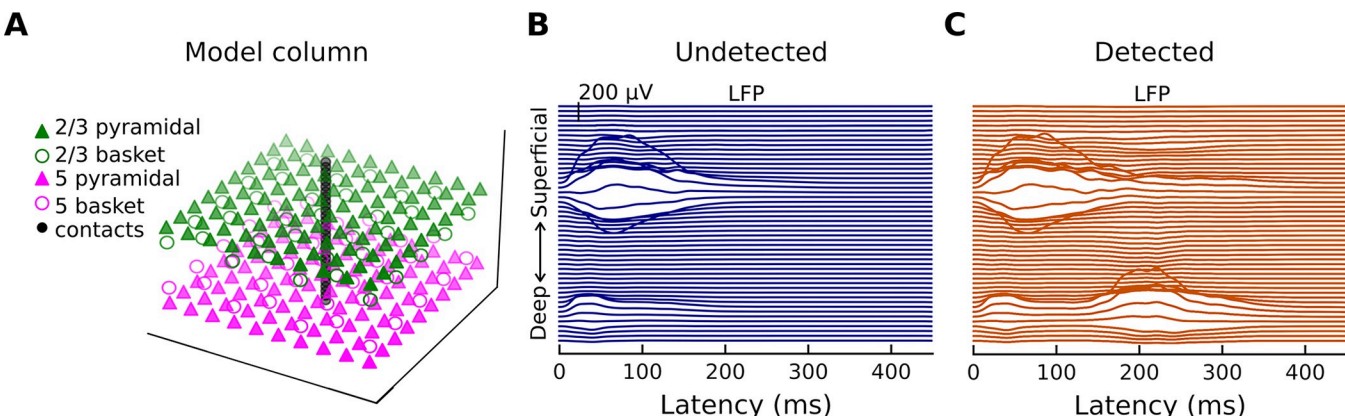

**Fig 5. Model cortical column and Local Field Potentials.** (**A**) HNN provides a model cortical column that consists of a scalable, ten-by-ten pyramidal neuron and basket cell grid (the ratio of pyramidal-neuron-to-basket-cell is 3-to-1). The LFP are estimated from a linear multi-electrode array of 50 electrode contacts. (**B**) Simulated LFP corresponding to one of the ten undetected target tone responses. (**C**) Simulated LFP corresponding to one of the ten detected target tone responses.

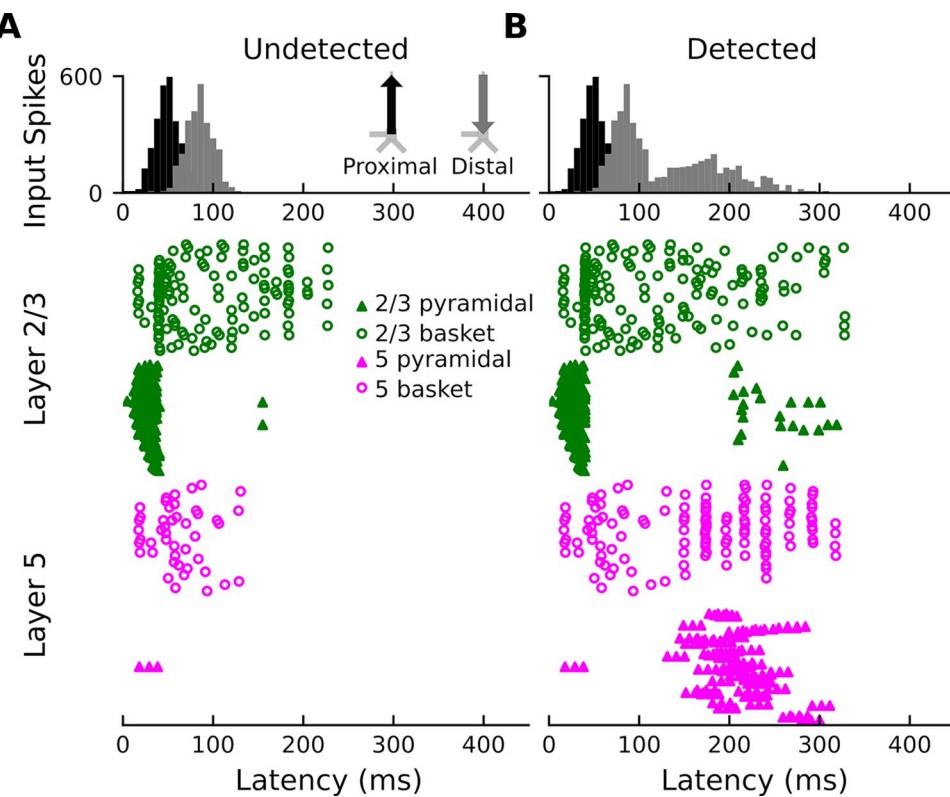

**Fig 6. Network spiking activity.** (**A**) Network spiking activity for undetected target tones. (**B**) Network spiking activity for detected target tones.

thalamo-cortical feedback loops, and the known importance of L5 pyramidal neurons in both auditory [40] and tactile [4,5] perception, a distal input (which conveys information via either feedback from higher cortical areas or thalamo-cortical-subcortical interactions), is appropriate. Nevertheless, we wanted to know whether the AAN could be accounted for via other circuit mechanisms. We investigated three alternative models: (1) perisomatic inhibition, (2) a reduced input sequence, and (3) an input sequence that has been previously used to simulate tactile [31] and auditory [24] evoked responses.

## Perisomatic inhibition

Perisomatic inhibition is mostly achieved by GABA$_A$ and GABA$_B$-releasing inhibitory cells. Of the two, GABA$_B$ receptors produce slower and more prolonged inhibition. To drive current down the pyramidal neuron dendrites via perisomatic inhibition with a timescale that matched the AAN, we modified the canonical circuit model by increasing the conductance of the GABA$_B$ receptors on the perisomatic compartment of L5 pyramidal neurons. The L5 basket cells thus had a stronger inhibitory effect on the somata of L5 pyramidal neurons (Fig 2); if the L5 basket cells are engaged via a proximal input (distal inputs do not target these cells), a stronger current source would form at the perisomatic compartment of L5 pyramidal neurons, all else being equal. After modifying the circuit, we introduced a second proximal input with decreased excitatory parameters and increased inhibitory parameters (S1–S4 Figs). This resulted in driving current down the L5 pyramidal neuron dendrites and closely replicating the AAN. After manual tuning and optimization, the R$^2$ values between empirical and simulated data are 0.10 and 0.85 for the responses to the undetected and detected tones,

respectively. RMSE is 1.41 nAm and 0.80 nAm for the response to the undetected and detected tones, respectively.

However, there are at least three reasons why this perisomatic inhibition model is less parsimonious than our favored model. First, it does not provide as close a fit to our data. This can be seen in particular in the response to undetected target tones, which takes longer to return to baseline following the distal input at 84.3 ms, a direct consequence of increased $GABA_B$ conductances. Second, we lack physiological justification for increasing $GABA_B$ conductances in the first place; indeed, altered $GABA_B$ receptor function is associated with a variety of disorders [41]. Third, and most importantly, increasing the effects of $GABA_B$ on the network resulted in the complete absence of L5 pyramidal neuron spiking, which we know is present and plays a critical role in perceptual detection of sensory stimuli [4,5].

### Reduced input sequence

The second alternative model we tried utilized a reduced input sequence (S5–S8 Figs). To model the response to the undetected target tones, we used a single proximal input; to model the response to detected target tones, we kept the proximal input fixed and introduced a single distal input. The $R^2$ values between empirical and simulated data are 0.08 and 0.90 for the response to the undetected and detected tones, respectively. RMSE is 0.61 nAm and 0.7 nAm for the response to the undetected and detected tones, respectively. While this model yielded a better fit to our data than did the perisomatic inhibition model, it did not match as well as our preferred model and did not account for the early negativity (~100 ms) that is present in response to both detected and undetected tones. Furthermore, spiking of L5 pyramidal neurons was observed quite early in this model, which does not fit with empirical data from other studies [4,5].

### Proximal-Distal-Proximal input sequence

Finally, we modified the reduced-sequence model to model the response to detected tones via an additional proximal input (S5–S7 and S9 Figs). The resulting proximal-distal-proximal input sequence has been used previously to model both tactile [31] and auditory [24] evoked responses. In this case, the response to the undetected tones is the same as in the reduced input sequence model, which means that the problem of underfitting persists. The $R^2$ value between empirical and simulated data for the response to the detected tones is 0.89 (RMSE is 0.73 nAm), so the additional proximal input did not improve the model fit over either the simpler proximal-distal input sequence or our preferred proximal-distal-distal input sequence. Thus, of all models, our preferred model provides the best fit to our data and is the most parsimonious, with as few changes to the inputs/parameters between conditions as possible while also being consistent with results from relevant animal models.

## Discussion

We used biophysical modeling (HNN) to better understand the cellular- and circuit-level dynamics of the cortical processing of target sounds in noise. Our model was able to reproduce the main features of an MEG-recorded response—the AAN—associated with perception (and lack thereof) of target tones embedded in multitone maskers [25,26]. We found that the AAN could be accounted for by additional input from the non-lemniscal thalamus or cortico-cortical projections to the distal apical dendrites of layer 2/3 and 5 pyramidal neurons. The input led to a prolonged negativity that closely resembled the AAN and was accompanied by both layer 5 LFP activity and large, burst-like spiking activity L5 pyramidal neurons.

## Model dynamics

In the case of the response to the undetected tones, the proximal input at 47 ms has the effect of momentarily driving spiking in L2/3 pyramidal neurons, with little corresponding activity in L5 pyramidal neurons (Fig 6A). This likely reflects model parameters, specifically differences in AMPA and NMDA conductances between L2/3 and L5 pyramidal neurons. In other words, the proximal input to the basal/oblique dendrites of L2/3 and L5 pyramidal neurons is defined by stronger L2/3 than L5 AMPA/NMDA conductances (Fig 4) which results in driving current up the dendrites in layer 2/3 (Fig 3C). Note, also, that part of the upward current in L2/3 pyramidal neurons could be a result of back-propagation of the spikes in those neurons (Fig 6A, solid green triangles) up their apical dendrites. At ~60 ms, somatic inhibition of L2/3 pyramidal neurons, and somatic and apical inhibition of L5 pyramidal neurons takes over (Fig 6A). This halts firing in both L2/3 and L5, returning the L2/3 current back to baseline, and driving weak, but significant current down the L5 pyramidal neuron dendrites. Current is further driven down the L5 pyramidal neuron dendrites as the result of the distal input (84 ms) to the apical dendrites of L2/3 and 5 pyramidal neurons.

In the case of the response to the detected tones, the distal input at 169 ms exhibits a much wider temporal distribution. This input, and its interaction with the previous inputs, induces several changes in the network dynamics whose net effect is to drive current downward in both L2/3 and, to a larger extent, L5 pyramidal neurons (Fig 3D). The slight undershoot of L2/3 dendritic currents at ~200 ms (Fig 3D) reflects the effects of the distal input on L2/3 pyramidal neurons and is likely the result of L2/3 pyramidal neuron firing at around the same time (Fig 6B). The distal input's most significant contribution, however, is the prominent, downward current flow in the dendrites of L5 pyramidal neurons (Fig 3D) which correlates with strong, bursting-like spiking activity of L5 pyramidal neurons (Fig 6B). The strength of NMDA receptor conductances (cf. Fig 4) in layer 5, as well as their inherent dynamics (i.e., slower time constants in NMDA vs. AMPA receptors) may be particularly important for this process. Additionally, the model column employed here, which uses the same local connectivity parameters as prior auditory work [24], accounts for realistic calcium dynamics that were updated from the original model column [14]. The firing of L5 pyramidal neurons leads to an influx of $Ca^{2+}$ that enters the cells at the distal compartments and diffuses down the neuron towards the soma, further driving current flow down the dendrites.

## The model is consistent with cellular- and circuit-level theories of conscious perception

Our modeling results are consistent with recent cellular- and circuit-level theories of consciousness / conscious perception, namely dendritic integration theory (DIT) [6,7] and apical amplification theory (AA) [8]. Both DIT and AA argue for (i) the importance of active $Ca^{2+}$ processes in the apical dendritic tufts of L5 pyramidal neurons, (ii) a multi-compartment L5 pyramidal neuron model, (iii) the fact that ascending sensory information gets amplified at the perisomatic output of L5 pyramidal neurons via conversion from single $Na^+$ spikes to burst spiking, and (vi) the potential to provide a cellular mechanism for network-level theories of consciousness / conscious perception (e.g., global workspace theory [42,43], recurrent processing theory [44,45]). In both models, conscious perception arises from the interaction between the sensory-specific input to the perisomatic compartment and the *amplificatory* effects of contextually relevant inputs to the apical compartment. If the apical integration zone receives sufficient excitatory input (from several possible sources), $Ca^{2+}$ spikes are initiated, amplifying the $Na^+$ spiking output of the cell (Fig 6B) and increasing the downstream effects on higher-order areas.

However, DIT and AA do differ in a few respects. First, the aspects of consciousness that they stress are different (states for DIT; perceptual content for AA). Second, what, specifically, gets amplified may differ (pure ascending input for AA vs. a match between ascending and descending input for DIT). And third, they emphasize different numbers of critical compartments in their respective L5 pyramidal neuron models (3 for DIT; 2 for AA). Specifically, in addition to the apical and perisomatic compartments argued for by both models, DIT argues for a coupling compartment that resides and permits coupling between the apical and perisomatic compartments and therefore of cortico-cortical and thalamo-cortical information loops [6]. In the context of the present study, because the subjects were conscious during the task, the coupling compartment can be assumed to be engaged, a necessary condition for conscious perception in DIT (although not *sufficient* for conscious perception according to either DIT or AA).

Considering both DIT and AA, we propose the following: First, information about the auditory stimulus arrives at the auditory lemniscal thalamus. This information is relayed to the basal/oblique dendrites of L2/3 and L5 pyramidal neurons in AC (modeled by the proximal input at 47.8 ms), where it causes current flow up the apical dendrites of L2/3 pyramidal neurons, resulting in the auditory evoked P1. Local cortico-cortical processing results in the first distal input that targets L2/3 pyramidal neurons (and L5 pyramidal neurons to a lesser extent based on model parameters) and generates the auditory evoked N1. At this point, the cortico-cortical loop is complete, but the fact that the N1 is present for both detected and undetected targets suggests that the type of cortico-cortical processing that generates the auditory N1 is not sufficient for conscious perception. For the undetected tones, no further cellular activity is detected. For the detected tones, information then propagates to the non-lemniscal thalamus where it is made available to higher cortical areas via the thalamo-cortical system. The non-lemniscal thalamus also projects back to the apical compartment of the local sensory circuit. The second distal input to the apical compartment arrives from either the non-lemniscal thalamus or cortico-cortical feedback, or both, and its interaction with the previous inputs results in the AAN. At this point, both AA and DIT predict that the stimulus is consciously perceived.

The timing of the second distal input, which is approximately 100–150 ms beyond the initial cortical representation, is consistent with the timing of the PAN, an evoked response thought to embody conscious perception across sensory modalities [28–30,32]. DIT and AA account for this latency by the time it takes for the non-lemniscal thalamus to extract information from higher cortical areas via thalamo-cortical loops before projecting back to the apical compartment. In the case of AA and DIT, the result is the contextualization and amplification of incoming sensory information that results in conscious perception. Although the cortical column model of HNN does not distinguish between L5a and L5b pyramidal neurons, based on prior animal work [5] as well as anatomical knowledge [46–48], the L5 activity of our model (and the AAN) may at least partially reflect activity of L5b pyramidal neurons. This case is made stronger by the fact that we incorporated updated $Ca^{2+}$ dynamics in our HNN model, which are known to be important in modeling the activity of L5b pyramidal neurons [49]. In the case of unconscious processing, DIT and AA predict that processing that does not involve L5 pyramidal neurons will remain non-conscious (reflected in our model by the lack of spiking of L5 pyramidal neurons in Fig 6A) because the thalamo-cortical loop is not engaged (in current context, the non-lemniscal thalamus interacts with the local cortical circuit primarily via L5b pyramidal neurons [7,8]).

## Relationship to network-level theories of conscious perception

It is important to note that neither our findings, nor AA or DIT, are inconsistent with *network-level*, neurobiological theories of consciousness [50,51] such as global workspace theory,

recurrent processing theory, predictive processing theory, and integrated information theory. Global workspace theory [42,43] proposes that for perceptual consciousness, the contents of consciousness must be accessible globally for memory, attention, action planning, etc. Recurrent processing theory [44,45], on the other hand, argues that conscious perception arises from a specific type of recurrent activity of cortico-cortical connections in sensory areas following initial feedforward information processing. Like recurrent processing theory, predictive processing theory [52,53], stemming from a predictive coding framework [54,55], relies heavily on feedback targeting the supragranular (and infragranular) layers. Finally, integrated information theory [56], which attempts to characterize the necessary properties of conscious systems (postulates) by first identifying the necessary properties of conscious experience (axioms), mainly argues that networks that support conscious experience must be highly interconnected and integrated, with some bias towards mechanisms located in sensory cortices. None of these theories attempts to describe what happens at the level of neurons nor cortical columns and our modeling work does not attempt to describe what takes place over large networks. Thus, both DIT and AA potentially provide the cellular-level ignition, recurrence, or integration mechanisms required by such network-level theories.

## Alternative models

Because this work is the result of computational modeling, which involves hand-tuning parameters over a large parameter space, we investigated several other approaches for modeling the AAN. These included a model with increased $GABA_B$ synaptic conductances, models with reduced input sequences (e.g., single proximal + single distal, single distal), and a model with an input sequence that has been used previously to simulate tactile and auditory evoked responses (proximal + distal + proximal). However, none of these other models were able to reproduce the specific peaks of our responses, particularly when considering responses to both detected and undetected targets. The model presented here came closest to reproducing all relevant waveforms while also accounting for relevant circuit structure and dynamics and being consistent with relevant animal models. For example, it is likely that the difference between the response to the detected and undetected target tones is due to a single ERF component that peaks at ~180 ms (Fig 1E). For this reason, the parameters of the first two inputs (used to model the response to undetected target tones) were kept constant and a single additional input was used to model the AAN. The difference waveform also revealed that the difference between the responses started at ~60 ms and lasted until ~360 ms. This indicated that a large standard deviation of the input time was needed to model the AAN. Importantly, it was the interaction of all three inputs that accounted for the AAN; a model with only the second of the two distal inputs did not provide a good fit.

## Relation to other paradigms

A common paradigm used to study auditory conscious perception is auditory streaming, in which an ABA pattern is presented and the subject reports whether the tones are perceived as a single stream (integration) or as two separate streams (segregation) [57]. A hallmark of this paradigm is that either spontaneously or with effort, a subject's perception switches between the two alternatives—a phenomenon called perceptual bistability [58–62]—but can be biased towards either. Our model is in accordance with basic principles of auditory streaming and perceptual bistability in that it potentially provides the neural mechanisms that could enable a subject to bias their perception via selective attention [62,63]. When subjects are asked to allocate their attention to either the integrated or segregated percept, they report hearing the associated percept for longer periods of time, and their brain responses resemble the patterns of

neural activity elicited by the same percept during passive listening [64]. If by the time the sensory-specific input (the ABA stream) arrives at the perisomatic compartment the subject's intention to bias their own perception has been established, then top-down input to the circuit could act to amplify—via synaptic gain, for example [65]—the desired stimulus features, thus increasing the dominance of the intended percept. The fact that subjects have voluntary control over which percept is experienced shows that information present in sensory inputs can be contextually modulated at the competition stage, which likely increases its effect on downstream targets. Although we do not explicitly model the effects of local inhibitory feedback on the AAN, it very likely impacts perception and selective attention in the multitone masking paradigm, as it does in streaming [58–60], via lateral inhibition of neurons selective for frequencies neighboring target tones embedded in the masker.

Although appropriate comparisons can be made between our model findings and perceptual awareness negativities in other sensory modalities, other negativities of similar latencies and scalp distributions like the early right anterior negativity (150–250 ms) [66], mismatch negativity (100–250 ms) [67–69], N2ac (100–400 ms) [70,71], and processing negativity [72–74] are not as straightforward. First, although the scalp distributions are superficially similar, each of the early right anterior negativity, MMN, and N2ac have slightly more anterior scalp distributions (with the early right anterior negativity weighting more towards the right hemisphere). Second, even if scalp distributions are similar, the function they reflect in perception and cognition and therefore the circuits and mechanisms giving rise to each may differ. For example, work from animal models suggests that the MMN may rely more heavily on mechanisms in layer 2/3 [69], as opposed to the importance of layer 5 for the AAN as suggested here. Of the responses listed above, the most likely to have circuit-level mechanisms similar to those underlying the AAN are the N2ac and processing negativity, by virtue of the fact that all three are likely related to attention. In any case, translating the mechanisms modeled here to other responses with similar spatiotemporal and—in the case of the N2ac and processing negativity—functional profiles will require further work.

### Future work

Insight into the cellular- and circuit-level activity that gives rise to consciousness can illuminate avenues for the diagnosis and treatment of multiple brain disorders and diseases, cognitive impairment, and psychiatric disorders. We believe that this study yields important constraints to the development of theories about the origins of conscious perception both within and beyond the auditory domain. Finally, although our model provides computational support for the importance of L5 pyramidal neurons in conscious perception (which has also been shown in experimental animals), empirical evidence for this is lacking, and further work is needed to test the model predictions. This can possibly be achieved in humans using recently developed human neuroimaging techniques including laminar M/EEG [75–77] and ultra high-field fMRI [16,17,78,79]. Counterpart methods such as current source density may permit us to test this with appropriate tasks in experimental animal models [80,81].

## Materials and methods

### Data

All data used for this modeling study came from prior work that examined MEG responses to target-tone streams embedded in random, multitone maskers [26]. Details of the experimental design, acoustic stimulation, task, and data analysis can be found there. We provide brief overviews below.

## Auditory stimulation

Participants (N = 20) were presented with two blocks of auditory oddball sequences. Each sequence (5-s long) contained ten short, evenly spaced pure tones (duration: 100-ms; stimulus onset asynchrony: 500 ms) at suprathreshold sensation levels. Nine of the tones were standard tones while one was a deviant tone that differed in frequency from standard tones by 1/12 of an octave in either direction. These oddball sequences were embedded in random multitone masking clouds, which are known to cause large amounts of informational masking [25,82]. Subjects were asked to indicate via a button press the moment at which they began to hear out the sequence of pure tones from the random multitone background and to ignore the deviant tones. Only the responses to the standard tones were modeled; the responses to the deviant tone were not relevant for the present study and therefore not modeled.

## MEG acquisition and analysis

The MEG responses were acquired using a Neuromag 122 system with orthogonal pairs of planar gradiometers at each of 61 locations around the head. MEG was acquired at a sampling rate of 500 Hz with a 160-Hz online lowpass filter. The data were then bandpass-filtered (0.5–15 Hz) offline, and a PCA-based artifact-rejection algorithm [83] was used to remove eye blinks and saccades. The data was then epoched from -50-450 ms around the onset of the target tones and binned into one of two categories: detected target tones and undetected target tones. Because at least two stimuli must be heard before a subject is able to detect a sequence, responses to the two target tones immediately preceding the button press, as well as the responses to all target tones after the button press, were placed into the detected bin. The undetected bin included responses to standard tones that were not perceived by the listener (i.e., those target tones occurring more than two tones prior to the button press).

HNN requires M/EEG responses to be source-localized. In their source estimation, Dykstra and Gutschalk used a combination of anatomically constrained, noise-normalized minimum-norm estimates [84,85] (also known as dynamic statistical parametric mapping [86]) and sparse inverse estimates based on an L1/L2 mixed norm [87]. Individualized structural MRI scans, including T1-weighted magnetization-prepared rapid gradient echo (MPRAGE) and multiecho fast low-angle shot (FLASH) sequences, were acquired for each subject using a 3-T Siemens TIM Trio scanner. Individual source spaces and boundary element models were created, and the source estimates were obtained for each subject's left and right AC. This was all carried out using MNE-Python [88] and Freesurfer [89]. Alignment of the MRI and MEG coordinate frames was done in MNE-Python [90].

After source localization in individual subjects, the grand average of both hemispheres was calculated and modeled for each response (detected target tones vs. undetected target tones. These source-localized signals (units of ampere-meters, Am) correspond to the net longitudinal current flow within postsynaptic pyramidal neurons—also known as primary currents—and can be thought of as primary current dipoles (with a defined strength and direction). HNN works by simulating the primary currents given a set of inputs and corresponding parameters and outputs the primary current dipoles which can then be directly compared to the empirical, primary current dipoles.

## Model

We used the open-source, computational neural modeling software Human Neocortical Neurosolver (HNN, available at https://hnn.brown.edu/) to model the AAN. HNN is designed to model source-localized M/EEG activity from a single cortical area to develop hypotheses about the underlying circuitry that gives rise to ERFs of interest. The model's underlying local

network architecture and neuronal morphology are based primarily on animal studies of the mouse/rat somatosensory cortex and cat visual cortex, but were modified in light of human anatomical findings [14]. Furthermore, it is known from anatomical studies that primary sensory areas share canonical cortical circuit features, even across species, justifying the use of HNN's circuit architecture in sensory areas outside somatosensory cortex [14,38,47,91–96]. This is particularly true as it pertains to the sufficiency of the model in accounting for primary M/EEG currents, the main output of the model. Along these lines, HNN has been used previously to model the contralateral dominance and hemispheric lateralization of human auditory evoked responses [24].

HNN provides a canonical model of a reduced neocortical column that includes both excitatory pyramidal and inhibitory basket neurons that span the supragranular (layers 2/3) and infragranular (layer 5) layers -layers that contribute the most to activity recorded with M/EEG sensors outside the head. The granular layer (layer 4) is not modeled because its main function is to project activity directly to layers 2/3 and 5. The model neocortical column closely describes the cellular and circuit-level activity of a typical neocortical column and is based on biophysical principles of the primary currents that arise from pyramidal neurons. The model pyramidal neurons are interconnected (as well as connected to the basket cells) via excitatory glutamatergic (AMPA/NMDA) synapses, while the basket cells are interconnected (as well as connected to pyramidal neurons) via inhibitory GABAergic (GABA$_A$/GABA$_B$) synapses [97]. There is also all-to-all connectivity between pyramidal neurons. These local network connections are fixed but the strengths of the connections between cell types can be adjusted.

HNN's canonical model functions as a general template that can be activated via several inputs. The template consists of two layers with 100 pyramidal neurons each in a ratio of three-to-one pyramidal-neuron-to-basket-cell for a total of 200 pyramidal neurons and 70 basket cells (total of 270 cells). A scaling factor can be applied depending on the magnitude of the signal we want to simulate (ERFs which are typically in the order of 10–1000 nAm require tens of thousands of pyramidal neurons to have synchronous dendritic currents). The inputs can be understood as a series of synaptic inputs that give rise to an ERF and can be one of two types: proximal or distal. _Proximal_ (to the soma) inputs target the basal and oblique dendrites of pyramidal neurons as well as inhibitory interneurons in layers 2/3 and layer 5, via the granular layer, and reflect feedforward information from the lemniscal thalamus. This influx of ions generates a sink-source pair that drives currents in pyramidal neuron dendrites oriented perpendicular to the cortical surface. MEG measures the magnetic fields induced by the primary current dipole outside the head [12–14]. These primary currents are precisely what HNN simulates. _Distal_ (to the soma) inputs are inputs that arrive from cortico-cortical feedback or non-lemniscal thalamus and target the apical dendrites of layer 2/3 and 5 pyramidal neurons and the soma of L2/3 basket neurons. This drives current flow down the dendrites of pyramidal neurons and away from the brain surface. Distal inputs represent feedback processing. It is important to note that the structures from which layer-specific inputs are relayed (i.e., the lemniscal thalamus, non-lemniscal thalamus, and higher-level cortical areas) are not explicitly modeled. Further, the model makes no distinction between cortico-cortical feedback and the non-lemniscal thalamus when it comes to defining distal inputs.

Each input consists of a set of parameters that can be adjusted to simulate the ERF of interest. These include the mean and standard deviation of the start time (ms), the number of spikes (the number of inputs provided to each synapse), as well as the strength/weight of each synaptic connection (where a weight of 0 μS corresponds to the lack of a connection). When thinking about the set of parameters that define a proximal or distal input, it might be useful to refer to Fig 2. AMPA and NMDA receptors are both ionotropic receptors that often coexist at the same synapse and are activated by the neurotransmitter glutamate (the major excitatory

neurotransmitter in the central nervous system). Glutamate produces excitatory postsynaptic potentials. HNN allows us to independently adjust the weight (synaptic conductance) for the AMPA and NMDA receptors of the targeted neuron's dendrites. For example, when defining a proximal input, one must adjust the weight of AMPA receptors of L2/3 pyramidal neurons. In this case, a high conductance value is equivalent to increasing the path for current flow, allowing more positive ions (specifically $Na^+$ and $K^+$ ions) to flow into the target cells. From literature [29] we know that excitatory postsynaptic currents have both an AMPA- and an NMDA-receptor component. While the NMDA component exhibits slower rise and decay times (given NMDA's higher affinity for glutamate that results in the slower dissociation of glutamate from the receptor), the AMPA component exhibits a rapid onset and decay. Increasing the weight of AMPA receptors of L2/3 pyramidal neurons may therefore have the effect of increasing the flow of current up L2/3 pyramidal neuron dendrites, leading to a positive peak that is brief (as recorded by M/EEG sensors outside the head).

To simulate the response to undetected target tones, we first loaded the corresponding source-localized MEG data into the software. The polarity of the source-localized dipole waveform gives specific insight into the direction of current flow within the pyramidal neurons across the cortical layers. A positive polarity indicates current flow up the dendrites, toward the brain surface, while a negative polarity indicates current flow down the dendrites, away from the brain surface and towards the white matter surface. To model the undetected response, we activated the model cortical column with a proximal input (mean ± s.d. = 47.8 ± 13.2 ms) followed by a distal input (84.3 ± 15.1 ms). The same parameter values as used in a similar study [24] were initially set and gradually changed until the morphology of the simulated waveform resembled the morphology of the empirical data. We ran a total of ten simulations from a distribution to generate variability and averaged the traces to arrive at the simulated waveform. Once a close fit was attained, we used HNN's optimization tool which performs a wide search over the parameter space (within specified ranges of parameter values) to achieve a tighter fit. Afterwards, we modeled the detected response by introducing an additional distal input (169.3 ± 50.4 ms) whose interaction with the previous inputs accounted for the AAN. We optimized the parameters of the last input while keeping those of the first two inputs fixed. A simple linear regression analysis was used to assess the variation of the recorded responses that can be explained by the model responses. $R^2$ values are reported in the Results. The RMSE values between the empirical and simulated responses were also calculated to assess the fit. A scaling factor of 300 was applied to match the magnitude of the recorded data, meaning that a total of approximately 60,000 cells are thought to give rise to the recorded dipole. The list of parameters that can be used to recreate the traces can be found in Fig 4.

HNN's outputs include individual layer activity (laminar profiles), LFPs, and individual cell spiking activity. Specifically, the laminar profiles estimate the individual contributions of L2/3 and L5 pyramidal neurons to the net current dipoles, while the LFPs estimate the extracellular potentials as measured across the depth of the model cortical column via a simulated linear multielectrode array with 100-μm intercontact spacing and an extracellular conductivity constant of 0.3 S/m. The individual cell spiking activity plots describe how the model column might give rise to the aggregate signal at the cellular level. Putting the three together allows us to study the origins of the AAN at the level of individual cells and neural circuits. It is important to note that though there are likely multiple input sequences and parameter sets that may reproduce the morphology of the empirical waveform, the particular input sequence and corresponding parameter values presented here are sufficient to account for the relevant components of both waveforms and are consistent with prior work in animal models [4,5]. Future modeling work of auditory evoked responses could incorporate more detail of both the auditory cortical circuit as well as the pattern of thalamo-cortical connectivity [98].

## Alternative models

**Perisomatic inhibition.**   Before simulating the responses in this alternative model, we modified the canonical model circuit by increasing the synaptic conductances of $GABA_B$ receptors on the perisomatic compartment of L5 pyramidal neurons from 0.025 to 0.05 μS. This resulted in the underlying circuitry being different, so the parameters of the proximal (36 ms ± 25 ms) and first distal (84.3 ms ± 15.1 ms) input were once again hand-tuned to fit the simulated response to the recorded response to undetected target tones and then optimized. To model the response to detected target tones, we introduced an additional proximal input (169.3 ms ± 50.4 ms) with weak excitatory and strong inhibitory parameters (S4 Fig). Then, the response was optimized while keeping the first two inputs fixed.

**Reduced input sequence.**   The second alternative model we tried used a reduced input sequence for both detected and undetected target tones compared to our preferred model. To model the response to undetected target tones we used a single proximal input (47.8 ms ± 13.3 ms). As in our preferred model and the perisomatic inhibition model, we hand-tuned model parameters to get an initially satisfactory fit prior to optimization. To model the response to the detected tones, we introduced a single distal input (154 ms ± 55.1 ms) on top of the single proximal input and hand-tuned the parameters of this input while keeping the proximal input fixed, and then ran optimization.

**Proximal-Distal-Proximal input sequence.**   We modeled the response to the undetected tones using the same parameters as in the reduced input sequence model. To model the response to the detected tones, we introduced a single distal input (154 ms ± 55.1 ms) as well as an additional proximal input (395.4 ms ± 20 ms). The same parameters as in a previous study [24] were initially set and hand-tuned to achieve a close fit. Optimization was then run while keeping the first proximal input fixed.

## Supporting information

**S1 Fig. Current dipoles and laminar profiles for the perisomatic inhibition model.** (**A**) Model output (dark blue) and data (light blue) for undetected target tones. A proximal input (36 ms) followed by a distal input (84.3 ms) drive the network. $R^2$ between empirical and simulated data is 0.10, RMSE is 1.41 nAm. (**B**) Model output (dark orange) and data (light orange) for detected target tones. The same proximal and distal inputs used to model the response to undetected target tones, and an additional proximal input (169.3 ms), drive the network. $R^2$ between empirical and simulated data is 0.85, RMSE is 0.80 nAm. (**C-D**) Laminar profiles for the responses to undetected and detected target tones. The corresponding input parameter values are displayed in S4 Fig. The network from which the simulated dipole activity arises consists of 60,000 cells.
(TIF)

**S2 Fig. Spiking cell activity for the perisomatic inhibition model.** (**A**) Network spiking activity for undetected target tones. (**B**) Network spiking activity for detected target tones. Note in both cases the complete absence of spiking activity in L5 pyramidal neurons.
(TIF)

**S3 Fig. Local field potentials for the perisomatic inhibition model.** (**A**) Simulated LFP to undetected target tones. (**B**) Simulated LFP to detected target tones.
(TIF)

**S4 Fig. Input parameters for the perisomatic inhibition model.** The canonical circuit model was modified by increasing the conductance of the $GABA_B$ receptors on the perisomatic

compartment of L5 pyramidal neurons.
(TIF)

**S5 Fig. Current dipoles and laminar profiles for the reduced-input (A, B, D, E) and proximal-distal-proximal (C, F) input seq0.0uences.** (**A**) Model output (dark blue) and data (light blue) for undetected target tones. A single proximal input (47.8 ms) drives the network. $R^2$ between empirical and simulated data is 0.08, RMSE is 0.61 nAm. (**B**) Model output (dark orange) and data (light orange) for detected target tones for the reduced input sequence model. The same proximal input that was used to model the response to undetected target tones, and a distal input (154.0 ms) drive the network. $R^2$ between empirical and simulated data is 0.90, RMSE is 0.70 nAm. (**C**) Model output (dark orange) and data (light orange) for detected target tones for the proximal-distal-proximal model. The same proximal input that was used to model the response to undetected target tones, a distal input (154.0 ms), and an additional proximal input (395.4 ms) drive the network. $R^2$ between empirical and simulated data is 0.89, RMSE is 0.73 nAm. (**D-F**) Laminar profiles for the responses to the undetected and detected target tones. The corresponding input parameter values are displayed in S8 and S9 Figs. The network from which the simulated dipole activity arises consists of 60,000 cells.
(TIF)

**S6 Fig. Spiking cell activity for the reduced-input and proximal-distal-proximal sequences.** (**A**) Network spiking activity for undetected target tones. (**B**) Network spiking activity for detected target tones for the reduced input sequence. (**C**) Network spiking activity for the detected target tones for the proximal-distal-proximal input sequence.
(TIF)

**S7 Fig. Local field potentials for the reduced-input and proximal-distal-proximal sequences.** (**A**) Simulated LFP for undetected target tones. (**B**) Simulated LFP for detected target tones for the reduced input sequence. (**C**) Simulated LFP for detected target tones for the proximal-distal-proximal input sequence.
(TIF)

**S8 Fig. Input parameters for the reduced input sequence.** The model column used was the calcium model column that consists of a more biologically accurate distribution of $Ca^{2+}$ channels on L5 pyramidal neurons compared to HNN's original Jones 2007 model column.
(TIF)

**S9 Fig. Input parameters for the proximal-distal-proximal input sequence.** The model column used was the calcium model column that consists of a more biologically accurate distribution of $Ca^{2+}$ channels on L5 pyramidal neurons compared to HNN's original Jones 2007 model column.
(TIF)

## Acknowledgments

We thank Drs. Stephanie Jones, Alexander Gutschalk, James Bonaiuto, Mainak Jas, and Carmen Kohl for helpful comments and discussion.

## Author Contributions

**Conceptualization:** Carolina Fernandez Pujol, Elizabeth G. Blundon, Andrew R. Dykstra.

**Data curation:** Andrew R. Dykstra.

**Formal analysis:** Carolina Fernandez Pujol.

**Funding acquisition:** Andrew R. Dykstra.

**Methodology:** Carolina Fernandez Pujol.

**Project administration:** Andrew R. Dykstra.

**Software:** Carolina Fernandez Pujol.

**Supervision:** Elizabeth G. Blundon, Andrew R. Dykstra.

**Visualization:** Carolina Fernandez Pujol.

**Writing – original draft:** Carolina Fernandez Pujol, Andrew R. Dykstra.

**Writing – review & editing:** Carolina Fernandez Pujol, Elizabeth G. Blundon, Andrew R. Dykstra.

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
