## [Decision Letter · Decision Letter 0]

16 May 2023

Dear Dr. Dykstra,

Thank you very much for submitting your manuscript "Laminar Specificity of the Auditory Perceptual Awareness Negativity: A Biophysical Modeling Study" for consideration at PLOS Computational Biology. As with all papers reviewed by the journal, your manuscript was reviewed by members of the editorial board and by several independent reviewers. The reviewers appreciated the attention to an important topic. Based on the reviews, we are likely to accept this manuscript for publication, providing that you modify the manuscript according to the review recommendations.

Dear Carolina and Andrew,

Thank you for submitting your research to Plos Comp. Biol. As you will read the reviewers were mostly enthusiastic about your work but raised some minor issues that you will want to address. I also enjoyed reading your contribution. I suggest also adding a color code (or some other marking such as an arrow) to the spectrogram of figure 1 to shown the location of the target stimulus - probably obvious to you.

Best wishes,

Frederic Theunissen

Sincerely,

Frédéric E. Theunissen

Academic Editor

PLOS Computational Biology

Daniele Marinazzo

Section Editor

PLOS Computational Biology

Dear Carolina and Andrew,

Thank you for submitting your research to Plos Comp. As you will read the reviewers were mostly enthusiastic about your work but raised some minor issues that you will want to address. I also enjoyed reading your contribution. I suggest also adding a color code (or some other marking such as an arrow) to the spectrogram of figure 1 to shown the location of the target stimulus - probably obvious to you.

Best wishes,

Frederic Theunissen

Reviewer's Responses to Questions

**Comments to the Authors:**

Reviewer #1: This is a highly valuable and unique contribution to the literature on auditory awareness, and I have no doubt that it should be published. I only have a few minor suggestions below, though a fair bit of new writing could be needed to broaden the Introduction and Discussion.

Currently, the Introduction and Discussion are perhaps too narrowly focused on awareness negativity and particular cortical column level mechanisms. However, I wonder if a broader discussion of other types of brain activity and mechanisms would make for a more engaging and impactful paper by truly connecting different grains of understanding of the neural basis of sensory awareness. For example, there is existing modeling of auditory bistable perception by Rankin/Rinzel and Denham, and more recently by David Little et al. in this journal that include biologically realistic mechanisms. Including these in a broader discussion and if possible seeing if any of the concepts in those studies (e.g., inhibition of competing neural populations, predictive coding) relate to the ones in your study would add a lot of value in moving towards general principles, rather than just trying to explain individual paradigms like informational masking or stream segregation. For example, your model includes inhibitory inputs not surprisingly given their ubiquity in neural circuitry, and you may react that for informational masking, there are no competing representations like in a bistable paradigm. But I would counter that when listening for a target of a particular frequency, there will always be need to inhibit neighboring neural populations that are selective for stimuli of a different frequency. Likewise, your model includes feedback input, which is a key neural architecture for predictive coding models and could be a key concept to explain the generation of the awareness negativities. It is also a possible way to connect awareness negativities to other large responses of similar latencies such as mismatch negativity and ERAN (evoked in studies of music processing, see studies by S. Koelsch, and his review in Trends in cognitive sciences, 2019). And there are models of mismatch negativity generation (that rely on feedback/predictive coding and NMDA receptors) that might be interesting to discuss and compare to your model.

I would remove all or most of the abbreviations, e.g., PNs. This will improve readability by quite a bit.

Pg 8, there is a mention of the auditory P0 and P1, but I wonder if instead of P0 you are referring to the Pb response. P0 is very short latency and precedes even the Pa middle-latency response, so well before the P1.

Pg. 14, need references to support the statement that “altered GABAB receptor function is associated with a variety of disorders”

Pg. 17 “network-level theories of consciousness” Does your modeling relate at all to any of the key concepts in information integration theory? If so, it would be nice to point out how.

Signed Review,

Joel Snyder

Reviewer #2: This manuscript describes an interesting exploratory modeling study of the potential biophysical sources of the auditory awareness negativity (AAN). The model is fit to MEG data, and the results indicate that synaptic input to the supragranular layers of auditory cortex (AC) best explains the AAN waveforms. Some alternative alternatives to the primary version of the model are investigated, and while these can capture aspects of the AAN data fairly well, they fail to explain the AAN data as fully as the primary model.

Overall the manuscript is well written, with fairly good English usage. In a number of place contractions are used (e.g., "they’re" on page 5, "doesn’t" on pages 14 & 15, and "didn’t" on page 15). It is preferable to avoid use of contractions in technical writing.

The modeling approach is well explained in the most part, and there is fairly good discussion and interpretation of the results. Two areas that could do with improved clarity are:

1. There is extensive discussion of currents flowing up and down dendrites starting on page 8, but it is only on page 11 that it is stated that this is referring to intracellular longitudinal current. It would be helpful for readers not familiar with the HNN modeling framework to state this explicitly on page 8, and perhaps to give a brief explanation of how these longitudinal intracellular currents provide predictions of the dipole sources generating the extracellular LFPs that are measured in MEG (cf. Fig. 1 of Neymotin et al. 2020 [14]).

2. RSME values for the model fits to the data are given without units. I believe from Fig. 3 that the units should be nAm - these units should be given after all RSME values. Additionally, it might be beneficial to also state the RMSE values as a percentage of the amplitude (peak-to-peak, say) of the empirical waveforms, to help the reader appreciate the relative size of the error. Finally, the labels on panels A and B of Figs. 3, S1, and S5 should be "RMSE", rather than just "RMS".

**Have the authors made all data and (if applicable) computational code underlying the findings in their manuscript fully available?**

Reviewer #1: **No: **They only offer data upon request. I would ask them to post the data and code on OSF or another repository immediately upon publication.

Reviewer #2: **No: **The authors state that the data and software are available upon request.

PLOS authors have the option to publish the peer review history of their article (what does this mean?). If published, this will include your full peer review and any attached files.

Reviewer #1: **Yes: **Joel Snyder

Reviewer #2: No

Figure Files:

Data Requirements:

Reproducibility:

References:

---

## [Editor Report · Decision Letter 1]

17 Jun 2023

Dear Dr. Dykstra,

We are pleased to inform you that your manuscript 'Laminar Specificity of the Auditory Perceptual Awareness Negativity: A Biophysical Modeling Study' has been provisionally accepted for publication in PLOS Computational Biology.

Best regards,

Frédéric E. Theunissen

Academic Editor

PLOS Computational Biology

Daniele Marinazzo

Section Editor

PLOS Computational Biology

Thank you for addressing all minor issues that were raised.

Regards,

Frederic Theunissen

---

## [Editor Report · Acceptance letter]

26 Jun 2023

PCOMPBIOL-D-23-00358R1 

Laminar Specificity of the Auditory Perceptual Awareness Negativity: A Biophysical Modeling Study

Dear Dr Dykstra,

I am pleased to inform you that your manuscript has been formally accepted for publication in PLOS Computational Biology. Your manuscript is now with our production department and you will be notified of the publication date in due course.

With kind regards,

Anita Estes
